# Intratumoral Cellular Heterogeneity: Implications for Drug Resistance in Patients with Non-Small Cell Lung Cancer

**DOI:** 10.3390/cancers13092023

**Published:** 2021-04-22

**Authors:** Vanesa Gregorc, Chiara Lazzari, Mario Mandalá, Stefania Ippati, Alessandra Bulotta, Maria Giulia Cangi, Abdelrahman Khater, Maria Grazia Viganò, Aurora Mirabile, Lorenza Pecciarini, Francesca Rita Ogliari, Gianluigi Arrigoni, Greta Grassini, Giulia Veronesi, Claudio Doglioni

**Affiliations:** 1Department of Oncology, IRCCS San Raffaele, 20132 Milan, Italy; chiara.lazzari@hsr.it (C.L.); ippati.stefania@hsr.it (S.I.); bulotta.alessandra@hsr.it (A.B.); vigano.mariagrazia@hsr.it (M.G.V.); mirabile.aurora@hsr.it (A.M.); ogliari.francesca@hsr.it (F.R.O.); 2Division of Pathological Anatomy, Papa Giovanni XXIII Hospital, 24100 Bergamo, Italy; mario.mandala@unipg.it; 3Unit of Medical Oncology, University of Perugia, 06123 Perugia, Italy; 4Pathology Unit, San Raffaele Scientific Institute, IRCCS, 20132 Milan, Italy; cangi.mariagiulia@hsr.it (M.G.C.); pecciarini.lorenza@hsr.it (L.P.); arrigoni.gianluigi@hsr.it (G.A.); grassini.greta@hsr.it (G.G.); doglioni.claudio@hsr.it (C.D.); 5San Raffaele Hospital, IRCCS, University Vita Salute, 20132 Milan, Italy; a.khater@studenti.unisr.it; 6Division of Thoracic Surgery, San Raffaele Scientific Institute, IRCCS, 20132 Milan, Italy; veronesi.giulia@hsr.it

**Keywords:** non-small cell lung cancer, NSCLC, tumor heterogeneity, genomic instability, targeted agents

## Abstract

**Simple Summary:**

The number of druggable tumor-specific molecular alterations in the treatment of non-small cell lung cancer (NSCLC) has grown significantly in the past decade. Emerging technologies such as liquid biopsy and single-cell methods allow for studying targetable drivers and develop personalized treatments. However, although new therapies confer prolonged disease control and high tumor response rates, most patients eventually progress on targeted treatments. Intratumoral heterogeneity is a frequent event in NSCLC, driving the tumor cells to develop adaptive or new resistance mechanisms within the drug environment. This review summarizes the current and upcoming research on the biological role of tumor heterogeneity, highlighting the link between early and acquired drug resistance and tumoral heterogeneity in targetable driver mutated NSCLC.

**Abstract:**

Tailored therapies based on the identification of molecular targets currently represent a well-established therapeutic scenario in the treatment of non-small cell lung cancer (NSCLC) patients. However, while aiming to improve patients’ response to therapy, development of resistance is frequently observed in daily clinical practice. Intratumoral heterogeneity is a frequent event in NSCLC, responsible for several critical issues in patients’ diagnosis and treatment. Advances in single-cell sequencing technologies have allowed in-depth profiling of tumors and attributed intratumoral heterogeneity to genetic, epigenetic, and protein modification driven diversities within cancer cell populations. This review highlights current research on the biological role of tumor heterogeneity and its impact on the development of acquired resistance in NSCLC patients.

## 1. Introduction

The genomic instability of tumors favors the onset of stochastic mutations in oncogenes and tumor suppressors, thus promoting heterogeneity and contributing to cancer evolution. Not all molecular alterations observed are drivers that are responsible for cancer cell survival [1,2]. In most cases, they are passengers that do not promote cancer progression. In cancer cells, clonal and subclonal mutations coexist. The former occur early in the tumorigenesis process, and represent the trunk of a tumor’s phylogenetic tree; the latter appear later, are responsible for progression, and are observed on the branches. In 96% of cases, the molecular alterations that develop in driver genes are clonal.

Genetic differences have been found between the primary tumors and the metastatic lesions [3], even though 95% of the mutations observed in driver genes are detected both in the primary and metastatic sites. Profiling of samples from patients with renal cell carcinoma [4] has shown differences in terms of mutations and chromosomal imbalances between the primary tumor and the metastases, thus confirming the existence of intratumor heterogeneity. Moreover, DNA copy number assessments have identified significant inter-tumor heterogeneity of brain metastasis (BM) compared to the primary lung cancer lesion in non-small cell lung cancer (NSCLC) [5,6]. Several mechanisms favor the development of tumor heterogeneity, including the proliferation of stem cells within tumors, genomic and chromosomal instability, epigenetic modifications, and the activation of mechanisms in response to a microenvironment [7].

The introduction of targeted agents in the therapeutic armamentarium of patients with cancer has significantly improved their rates of survival [8,9,10]. However, despite the progress observed with precision medicine approaches and the high objective response rate observed in molecularly defined subgroups of patients with tumors, complete responses are uncommon, and progression of residual disease occurs [11]. Tumor heterogeneity represents the molecular basis responsible for primary and acquired drug resistance [12].

In the current review, we will provide an overview of the role of tumor heterogeneity in patients with NSCLC, a disease in which the molecular characterization has become an essential requirement for defining the treatment strategy. The practice of performing re-biopsies following progression has helped to molecularly characterize the population of quiescent drug-tolerant cells and to identify the alternative pathways and survival signals activated. Moreover, the use of next generation sequencing (NGS) to perform comprehensive molecular characterization has significantly contributed to our understanding of the molecular basis of intratumor heterogeneity and better define patient prognosis.

This review will focus on the differences in molecularly defined subgroups of patients with NSCLC and how these differences influence treatment outcomes and patient survival. In particular, we will discuss KRAS and EGFR mutated, as well as EML4-ALK-rearranged, NSCLC, in which intratumoral heterogeneity has been studied as a mechanism of resistance to treatments.

## 2. Heterogeneity in Histology in NSCLC

Genome-wide sequencing analyses have revealed an average of approximately 150 somatic mutations in patients with lung cancer [13]. Lung cancer is one of the tumors with the highest frequencies of non-synonymous mutations [14], probably due to the carcinogenic stimulus induced by tobacco. NSCLC, which accounts for approximately 80% of lung tumors, is a heterogeneous disease at the histological and molecular level [15]. NSCLC is classified into different histologies, the most common being epidermoid carcinomas and adenocarcinomas, followed by large cell carcinomas, large cell neuroendocrine carcinomas, adenosquamous carcinomas, squamous carcinomas, and sarcomatoid and pleomorphic carcinomas. This heterogeneity is the result of the different cells of origin and the different molecular pathways activated [16].

Squamous cell carcinomas arise from the basal cells located in the respiratory epithelium, and are generally observed in smokers. The sequence of events inducing the development of a squamous tumor includes the onset of hyperplasia in basal cells, followed by metaplasia, dysplasia, and finally resulting in a squamous carcinoma. Specific molecular alterations accumulate during each step of the tumorigenesis, and lead to the growth of more aggressive lesions [17]. Mutations in PI3KCA, SOX2, CDK2, P63, PTEN, Rb1, CDKN2A and DDR2 genes, along with amplification of the FGFR1 and MET genes, are frequently detected in patients with squamous lung tumors [16,18,19].

Lung adenocarcinomas may develop in smokers and non-smokers, but tobacco as a risk factor can affect tumor biology. Adenocarcinomas originate from the cells located within the terminal respiratory unit, where the expression of markers such as CK7 and TTF1 are generally found. In the case of less differentiated lung adenocarcinomas, CK20 or markers from colic-like origin may be detected. The type of the cells of origin, the grade of differentiation, and the molecular alterations emerging in the course of carcinogenesis result in the generation of different subtypes of lung adenocarcinoma. Three patterns are observed according to the cellular organization: the papillary, the solid, and another including the acinar, the lepidic, and the micropapillary [20], as reported in Table 1 and Figure 1a,b. Figure 1 also shows the differences between tissue (c) and liquid biopsy (d). Lower TTF1 expression is observed in those tumors with papillary or solid patterns compared with the acinar, the lepidic, and the micropapillary. Higher levels of HER2 and lower ERCC1 expression are found in the papillary subtype, where defects in the DNA repair system have a pathogenetic role. Conversely, lower HER2 and higher EGFR and ERCC1 expression are detected in those adenocarcinomas with a solid pattern, which are less differentiated tumors.

Thanks to the use of NGS panels, concurrent molecular alterations have been described in molecularly selected classified lung tumors, thus further increasing the complexity of NSCLC biology. These molecular alterations contribute to the emergence of tumor heterogeneity, which in turn impacts patient prognosis and sensitivity to treatment.

## 3. Heterogeneity in KRAS Mutated NSCLC Patients

Mutations in KRAS are the most frequently observed molecular alterations in patients with lung adenocarcinoma, accounting for approximately 30% of cases [21]. They are located in the GTP-binding protein (GAP), where their presence inhibits guanosine-5’-triphosphate (GTP) hydrolysis. As a consequence, the constitutive activation of the downstream pathways occurs [22]. KRAS mutations are observed in codons 12, 13, and 61. According to the amino acid substitutions, different intracellular signaling pathways are activated, thus influencing the pathogenesis of KRAS mutant NSCLC [23].

KRAS mutant NSCLCs are heterogeneous tumors, due to the presence of co-occurring alterations in genes other than KRAS [24]. Based on transcriptional, mutational, copy number, and proteomic data, KRAS mutated NSCLC patients are classified into three groups: one, including co-mutations in *TP53* (KP); a second carrying inactivating mutations in the tumor suppressor liver kinase b1 (LKB1) (KL); and a third, with bi-allelic deletions of two tumor suppressor genes, CDKN2A and CDKN2B (KC) [24]. Within the three subgroups, no correlation has been observed with the KRAS mutant alleles. However, a correlation has been found between patient prognosis and response to treatment. Patients included in the KP group show a higher mutational burden, increased expression of genes involved in the immunological response, and increased activation of the JAK-STAT pathway compared to the other subtypes’ patients.

Conversely, in those harboring LKB1 mutations, the immune response is negatively affected, while in those carrying CDKN2A or CDKN2B alterations, low levels of TTF1, with a high expression of markers of mucinous differentiation, have been detected. Results from a retrospective analysis performed in patients with KRAS mutant NSCLC receiving immune checkpoint inhibitors demonstrated a significantly lower response rate (7.4% vs. 35.7%) and shorter progression free survival (PFS) and overall survival (OS) in those patients included in the KL subgroup compared with those in the KP subgroup [25]. These results were confirmed in a further retrospective analysis performed in KRAS mutant patients enrolled in the phase III CheckMate 057 trial, comparing nivolumab with docetaxel. Whole exome sequencing and the study of the tumor microenvironment in surgically resected specimens of NSCLC patients demonstrated low PDL1 expression, and a low percentage of CD3+ and CD8+ T-lymphocytes in LKB1 mutated patients. Preclinical findings showed the hyper activation of the MEK/ERK pathway in mice carrying the KRAS mutation only and in those with concurrent TP53 mutation, while a higher stimulation of AKT and SRC was observed in those harboring concomitant LKB1 mutation. These genetic differences might have an impact on treatment outcome. Docetaxel alone or in combination with the MEK inhibitor selumetinib [26] resulted in lower efficacy in KRAS mutant mice harboring LKB1 or TP53 mutations, compared with those carrying KRAS mutations only. Conversely, selumetinib improved the response in those with TP53 mutations, but not in those with LKB1 alterations. LKB1 inactivating mutations are observed in approximately 30% of patients with lung adenocarcinoma, and more frequently in those with KRAS mutant NSCLC [27]. In a retrospective analysis, concomitant KRAS and LKB1 mutations were associated with worse patient prognosis [28]. However, novel therapeutic inhibitors for KRAS treatments are undergoing clinical trials [29]. Recently, sotorasib has been demonstrated to be effective in patients with NSCLC harboring KRAS G12C mutations [29]. A phase III study comparing sotorasib with docetaxel in previously treated patients with advanced NSCLC has recently been conducted.

The heterogeneous biology observed in KRAS mutant NSCLC partially explains the difficulties experienced in developing efficient therapies targeting the KRAS gene. These data suggest the importance of performing a comprehensive molecular classification in the clinical trials exploring new agents targeting KRAS mutations in order to establish their activity in molecularly defined subgroups, better define patient prognosis, and eventually develop combinatorial approaches.

## 4. Heterogeneity in EGFR Mutated NSCLC

Somatic activating mutations in the tyrosine kinase domain of the EGFR gene are observed in 15% of patients with lung adenocarcinoma, and more frequently in those with lepidic and acinar subtypes [30,31,32], with exon 19 deletion (62%) and L858R point mutation in exon 21 (38%) being the most common. Results from prospective clinical trials comparing EGFR-TKIs with platinum-doublet chemotherapy showed shorter PFS in patients carrying EGFR L858R exon 21 mutation compared with those harboring exon 19 deletion [33]. The molecular basis of this clinical behavior is partly dependent on the higher somatic mutation burden identified by NGS in patients with exon 21 mutation compared with those with exon 19 deletion [34]. These findings suggest that heterogeneous preexisting sub-clones emerge under the pressure of the treatment, thus contributing to the development of resistance mechanisms and the poor prognosis observed in patients with exon 21 mutation. However, the mechanism behind how existing clones influence the initial response to primary TKI treatment is not fully understood. Intrinsic resistance resulting in inefficacy of anti-EGFR TKIs has recently been associated with the presence of co-occurring non-sensitive EGFR mutations [35,36,37]. Moreover, the role of other cancer driver gene co-mutations in intrinsic resistance has been addressed in several studies [38].

Results from the sequencing of 200 patients with advanced EGFR mutated NSCLC identified co-occurring mutations in TP53, PIK3CA, catenin beta-1 (CTNNB1), and RB1 [39]. Several alterations in TP53 were found, although their functional role remains unclear. However, shorter OS and PFS were observed in patients with TP53 mutations, while loss of function mutations in TP53 and RB1, occurring in nine percent of EGFR mutated cases, is associated with an increased risk of transformation into small cell lung cancer (SCLC). Concomitant amplification of mesenchymal-epithelial transition factor (MET) or ERBB2 genes significantly reduces PFS in patients receiving epidermal growth factor receptor tyrosine kinase inhibitors (EGFR-TKIs) [39].

Tumor re-biopsies performed in EGFR mutated NSCLC patients at the time of progression to erlotinib, gefitinib, afatinib, or osimertinib have documented several mechanisms of acquired resistance [40,41,42,43], including transformation to SCLC, the emergence of secondary resistance mutations able to interfere with drug binding, the activation of alternative signaling pathways, and mesenchymal changes in the tumor phenotype (Table 2). EGFR T790M mutation in exon 20 is the most common acquired resistance mechanism to first and second generation EGFR-TKIs. However, EGFR T790M has also been documented on pretreatment tumor samples in a subgroup of patients [44], thus suggesting the existence of sub-clones before the beginning of treatment. Shorter PFS was reported in those patients with co-occuring EGFR sensitizing mutations and exon 20 EGFR T790M mutation. Although a baseline EGFR T790M mutation was not analyzed in the phase III FLAURA trial [45] comparing erlotinib or gefitinib with osimertinib, the significantly longer PFS and OS registered in the osimertinib patients might be partially explained by its effective inhibition of baseline EGFR T90M sub-clones.

Similarly to the molecular findings observed in patients progressing on gefitinib and erlotinib, results from tumor re-biopsy and NGS analysis performed on tissue DNA and circulating tumor DNA in patients progressing on osimertinib identified secondary mutations interfering with drug binding, the most frequent being EGFR C797S [41,42,43,46]. Moreover, the loss of EGFR T790M [47] and the activation of bypass signaling pathways, including MET amplification, PIK3CA, KRAS, BRAF mutations, RET [48], FGFR3, and BRAF and ALK fusions, were described. Longer responses were registered in those patients receiving osimertinib as second line treatment following progression to first-generation EGFR-TKIs who maintained EGFR T790M in the course of osimertinib compared with those in which EGFR T90M was lost. The loss of EGFR T790M suggests the emergence of alternative mechanisms of acquired resistance, as the result of a more heterogeneous tumor with a more aggressive phenotype.

In conclusion, the understating of the molecular basis of tumor heterogeneity in EGFR mutated NSCLC patients is crucial for designing combinatorial approaches targeting the bypass signaling pathways, preventing the onset of resistance mechanisms, and prolonging patient survival.

## 5. Heterogeneity in EML4-ALK Rearranged NSCLC

EML4-ALK translocations are observed in 5% of patients with lung adenocarcinoma, more frequently in those with the acinar, solid, or signet cell patterns [49]. EML4-ALK is a chimeric protein with oncogenic activity. It derives from a rearrangement occurring between the N-terminal portion of the EML4 gene and the tyrosine kinase domain of the ALK gene [50]. According to the exon where the EML4 breakpoint occurs (exons 2, 6, 13, 14, 15, 17, 18, and 20), at least 21 variants have been reported [51,52,53], the most common being variant 1 and variants 3a/b, consisting of exons 1–13 and 1–6 of EML4, respectively [49,54,55]. Additional fusion partners of ALK have been identified, including TGF, KLC1, and KIKF5B [56]. In all cases, the ALK truncation occurs at exon 20, and the portion of the ALK gene included in the fusions involves the region between exons 20–29. Although the oncogenic transformation of EML4-ALK rearrangement is the result of the constitutive activation of tyrosine kinase domain of the ALK gene, EML4 influences the protein stability and the activated downstream pathways. Variants 1 and 2 are less stable compared with variants 3a/b and 5, thus affecting their sensitivity to therapy [57,58]. Preclinical findings suggest that oncogenic EML4-ALK fusion activates ALK, STAT3, ERK, and AKT signaling [59].

Among these pathways, RAS seems to be the most commonly activated, thus contributing to the development of resistance. Conversely, in variant 3, RAS signaling is not activated, and alternative resistance mechanisms emerge.

As observed in KRAS and EGFR mutated NSCLC, in EML4-ALK positive patients, concurrent mutations in the TP53 gene have been reported. TP53 mutations have been associated with genomic instability and concomitant amplifications in different genes, including MYC, CCND1, TERT, BIRC2, ORAOV1, and YAP1 [60]. Moreover, a shorter duration of response to crizotinib and a poorer survival rate [61] were documented in this subgroup of patients, probably because the genetic instability led to a heterogeneous disease, that favor the onset of multiple acquired resistance mechanisms.

Three generations of ALK inhibitors have been developed during the last 10 years: crizotinib, ceritinib, alectinib, brigatinib, and lorlatinib. These compounds have been designed with the aim of overcoming the ALK secondary mutations that develop in the course of treatment with the previous generation of ALK inhibitors and of improving blood brain barrier penetration.

As observed in EGFR mutated patients, the resistance mechanisms to ALK inhibitors include the onset of secondary mutations in the tyrosine kinase domain of the ALK gene, the development of mechanisms that prevent drug binding, ALK amplification, and the activation of alternative pathways (Table 3). Different secondary mutations have been identified following treatment with crizotinib, ceritinib, alectinib, brigatinib, and lorlatinib. Consecutive tumor re-biopsies performed in patients receiving ALK inhibitors sequentially allowed the molecular basis for tumor evolution in EML4-ALK NSCLC to be determined [62]. Among 55 patients progressing on crizotinib, ALK secondary mutations were identified in 20% of cases, while in ceritinib resistant tumors and in alectinib resistant samples, secondary ALK mutations were observed in 54% and 53% of cases, respectively. Ninety-one percent of those patients receiving ceritinib, and 100% of those progressing on alectinib, had been previously treated with crizotinib as afirst-line treatment. These findings suggest that ALK secondary mutations accumulate over the course of treatment due to the proliferation of resistant sub-clones, thus favoring the development of more heterogeneous tumors. Among the secondary ALK mutations currently identified, L1196M, G1269A, S1206Y, F1145C, and S1206Y are associated with resistance to crizotinib, but with sensitivity to ceritinib and alectinib [63,64,65,66,67]. C1156Y and F1174L confer sensitivity to alectinib, but not to ceritinib, while I1171T is sensitive to ceritinib, but not to alectinib [68]. Finally, G1202R causes resistance to crizotinib, ceritinib and alectinib, but not to brigatinib and lorlatinib [69,70,71]. However, a recent report, identified primary resistance to brigatinib in NSCLC patients carrying an ALK G1202R mutation [72]. These data confirm the importance of performing tumor re-biopsy at the time of progression to define the molecular basis of resistance and define a personalized treatment strategy.

Alternative signaling pathways might emerge in the case of ALK inhibitors. EGFR, KRAS mutations, and KIT amplification were observed in patients progressing on crizotinib [73,74]. Preclinical data indicate the hyper activation of EGFR, IGF-1R, and HER3 in NSCLC cell lines with acquired resistance to ceritinib [75], and an increase in phosphorylation of EGFR, HER2 [68], and IGF-1R in NSCLC cell lines with acquired resistance to alectinib.

Comparable to EGFR mutated NSCLC, conversion from adenocarcinoma to SCLC has been documented in a patient progressing on alectinib [76], thus confirming that histology transformation represents an escape mechanism in molecularly defined subgroups of NSCLC patients receiving a targeted agent.

## 6. SCLC Transformation

Transformation to SCLC is observed in approximately 5–14% of patients with lung adenocarcinoma carrying EGFR activating mutations [77,78,79]. Histological conversion has also been observed in EML4-ALK positive patients and in patients with lung adenocarcinoma and no targetable molecular alterations [77]. The pathogenetic basis of this phenomenon is not fully understood yet, but tumor heterogeneity has a pivotal role. Phenotypical epithelial-mesenchymal transition was described in a 38-year old male harboring secondary ALK and p.V600E BRAF mutations [80].

Mutations in RB1 and TP53 genes also seem to play an important role. Results from longitudinal tumor biopsies in a subgroup of EGFR mutated patients indicate that inactivation of RB1 and TP53 occurs early in adenocarcinoma cells [81] and may involve the alveolar type II cells. These sub-clones coexist in a dormant state and proliferate slow. No molecular alterations known to induce resistance to EGFR-TKIs have been identified in these sub-clones, thus suggesting the divergent evolutionary nature of SCLC transformation. However, the apolipoprotein B mRNA editing catalytic polypeptide-like (APOBEC)-mutational signature, which confers genetic instability, has been observed. As a consequence, hypermutation favors the proliferation of SCLC sub-clones, the onset of mutations in PIK3CA, the amplification of the MYC gene, or the loss of heterozygosity of chromosomes 13 and 17. These events contribute to the development of aggressive tumors.

## 7. Discussion and Future Directions

Our understanding of the molecular basis of NSCLC has allowed us to identify specific subgroups of molecularly defined lung tumors. The development of drugs designed to specifically inhibit these pathways has significantly prolonged patient survival. However, despite the high overall response rate and increased PFS and OS, complete responses are rare, and acquired resistance mechanisms inevitably emerge, resulting in tumor progression. Intratumoral heterogeneity is a frequent event in NSCLC, and represents the molecular keystones for tumor recurrence.

The use of NGS and sequential re-biopsies in patients progressing on targeted therapies have significantly contributed, in some cases, to defining the temporal acquisition of mutations and shedding light on the molecular determinants of NSCLC heterogeneity. The advances in the comprehension of NSCLC pathogenesis have resulted in important clinical implications.

First, the identification of concomitant alterations in subgroups of molecularly classified patients has helped to better characterize NSCLC biology, define patient prognosis, and establish new therapeutic strategies. Interpreting NGS data and determining the biological function and impact of co-occurring alterations requires further study. The building of centralized databases, including clinical and molecular data, will help to address this issue. In the last 20 years, we have moved from a classification based on histology to a molecular classification, where the identification of molecular alterations in targetable driver genes has become helpful in guiding therapy. Now we have entered into an era in which we are learning about the clinical impact of molecularly sub-classifying patients. In this scenario, the interaction between tumor microenvironments and immune cells remains to be further addressed.

Among the strategies applied to target intratumor heterogeneity, local treatments, including surgery or radio surgery, have been used, especially in those patients with oligo-metastatic disease, with the aim of eliminating the quiescent surviving cells that are not responding to treatment and may be responsible for progression.

All these data suggest that the complexity of the entire tumor may not be represented by a single tissue biopsy. Methods able to molecularly characterize circulating tumor DNA (ctDNA) have been introduced in clinical practice. ctDNA shedding by certain cancer cell types, analyzed by NGS, might help to determine the genetic composition of heterogeneous cancer sub-clones. Similarly, the isolation of circulating tumor cells (CTC), and their culture to generate CTC-derived models might be useful for defining the mediators of resistance and design strategies to overcome them.

## Figures and Tables

**Figure 1 cancers-13-02023-f001:**
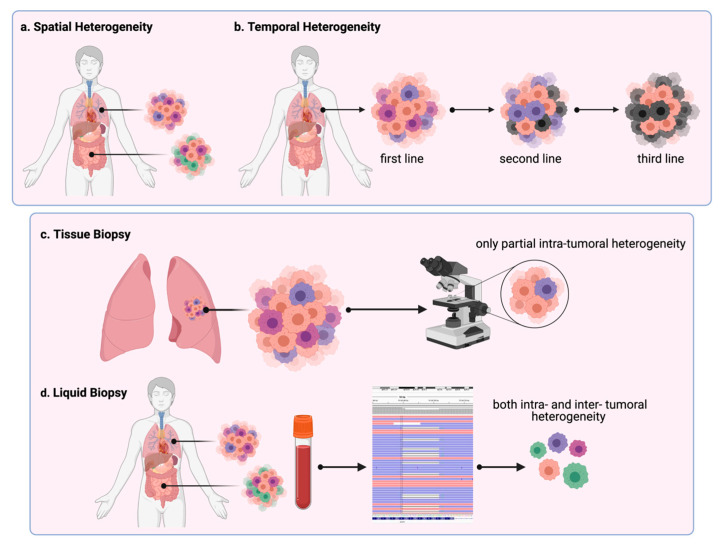
Tumor heterogeneity. Levels of heterogeneity: (**a**) different subclones in the primary tumor and metastasis define the spatial intra- and inter-tumoral heterogeneity; (**b**) differences among the primary tumor and tumor evolution after first, second, and third-line therapy and among metastases constitute temporal heterogeneity; (**c**) tissue biopsy performed on the primary tumor allows the assessment of only partial intra- and inter-tumoral heterogeneity; (**d**) liquid biopsy and sequencing allow detection of both the spatial and temporal tumor heterogeneity.

**Table 1 cancers-13-02023-t001:** Histologic patterns within lung adenocarcinoma.

PATTERN	IHC Markers
Papillary	low TTF1, high HER2, low ERCC1
Solid	low TTF1, low HER2, high ERCC1, high EGFR
Acinar, lepidic, micropapillary	high TTF1, high RB1

**Table 2 cancers-13-02023-t002:** Mechanisms of resistance to EGFR-TKIs.

Mechanism of Resistance			Drug Resistant
EGFR—dominant mechanisms		exon 20 EGFR T790M mutation	erlotinib gefitinib afatinib
		EGFR C797S	osimertinib
		G724S	
		L718Q	
EGFR—non-dominant mechanisms	bypass signaling pathways	MET amplification	erlotinib gefitinib afatinib osimertinib
	SCLC transformation	-	erlotinib gefitinib afatinib osimertinib
	activation of bypass signaling pathways (concomitant with loss of EGFR T790M)	PIK3CA, KRAS, BRAF mutations, RET, FGFR3, BRAF and ALK fusion	osimertinib

**Table 3 cancers-13-02023-t003:** Mechanisms of resistance to ALK inhibitors.

Mechanism of Resistance		Drug Resistant
ALK—dominant mechanisms	L1196M	crizotinib
	G1269A	crizotinib
	G120R	crizotinib, ceritinib, alectinib
	S1206Y	crizotinib
	L1152R	crizotinib
	F1147L	crizotinib, ceritinib
	F1245C	crizotinib
	I1171T (N)	alectinib
	F1174C(V)	ceritinib
	G1123S	ceritinib
	V1180L	alectinib
	L1198F + C1156Y	lorlatinib
ALK—non-dominant mechanisms	EGFR mutations	crizotinib
	KRAS mutations	
	KIT amplification	
	p EGFR	ceritinib
	IGF-1R	
	HER3	
	p EGFR	alectinib
	p HER2	
	IGF-1R	
	SCLC transformation	

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
