# Peer review of "Intratumoral Cellular Heterogeneity: Implications for Drug Resistance in Patients with Non-Small Cell Lung Cancer"

_cancers, 2021, doi:10.3390/cancers13092023_

Round 1
Reviewer 1 Report
As a result of advances in sequencing technologies such as next-generation sequencing (NGS), this has permitted comprehensive genomic analysis of tumours, in particular NSCLC, and the study of intratumor heterogeneity. The process of temporal and spatial changes within the cancer genome is now recognized as a molecular mechanisms underlying cancer heterogeneity. It is now well established that many different cancer types arise due to the accumulation of multiple genomic abnormalities, resulting in clones of tumor cells with specific genomic abnormalities.
In this review by Gregorc et al., the authors present a review on the implication of intratumour heterogeneity in the development of drug resistance in NSCLC patients and in doing so, explore the extent and clinical implications of this phenomenon. In particular, the authors discuss this in the context of molecularly defined tumour subsets such as KRAS, EGFR and EML4-ALK NSCLC and the transformation from adenocarcinoma histology to SCLC histology.
The review is well-written, concise yet comprehensive, with appropriate examples provided. A number of minor issues identified are provided below:
[1] In certain parts of the manuscript, there is a lack of appropriate referencing with respect to certain sections discussed. In order to further improve the quality of the review, it would be useful to supplement the current review with appropriate references supporting the relevant sections described. These include descriptive sections relating to the literature on lines 33-34; lines 84-85; lines 99-101; lines 136-138; lines 176-177 (SCLC transformation); lines 246-247; lines 277-278; lines 281-282; lines 287-288. Studies/findings stated by the authors in the above, should be backed up with appropriate references.
[2] Table 2 (Mechanisms of resistance to EGFR-TKIs) in its current form is slightly dis-jointed with the text under "mechanisms of resistance" not aligned and orientated correctly.
[3] On line 205, it currently states "EGFR T90M". This should be corrected to "EGFR T790M".
[4] Throughout the manuscript, the authors use the word "bases" instead of "basis". One example of this is on line 280..."The pathogenetic bases..." This should be corrected throughout.
[5] Somewhat similar to the above, the authors use the phrase "...progressing to crizotinib...alectinib" etc instead of "...progressing on crizotinib..." etc. This minor grammatical error should also be addressed throughout.
Author Response
PtP response cancers-1180866-Point-to-Point reponse to Reviewer 1
We thank the reviewer for the valuable comments and feedback on our manuscript. Please find our responses below.
[1] In certain parts of the manuscript, there is a lack of appropriate referencing with respect to certain sections discussed. In order to further improve the quality of the review, it would be useful to supplement the current review with appropriate references supporting the relevant sections described. These include descriptive sections relating to the literature on lines 33-34; lines 84-85; lines 99-101; lines 136-138; lines 176-177 (SCLC transformation); lines 246-247; lines 277-278; lines 281-282; lines 287-288. Studies/findings stated by the authors in the above, should be backed up with appropriate references.
- New references have been added throughout the text.
[2] Table 2 (Mechanisms of resistance to EGFR-TKIs) in its current form is slightly dis-jointed with the text under "mechanisms of resistance" not aligned and orientated correctly.
- The table has been fixed and aligned as suggested
[3] On line 205, it currently states "EGFR T90M". This should be corrected to "EGFR T790M".
- Resolved
[4] Throughout the manuscript, the authors use the word "bases" instead of "basis". One example of this is on line 280..."The pathogenetic bases..." This should be corrected throughout.
- The typo has been corrected throughout the text
[5] Somewhat similar to the above, the authors use the phrase "...progressing to crizotinib...alectinib" etc instead of "...progressing on crizotinib..." etc. This minor grammatical error should also be addressed throughout.
- The error has been corrected throughout the text
Reviewer 2 Report
Thank to dr Gregorc and colleagues for successfully performed the review article containing the most important factors in tumor heterogeneity, which is currently very debatable topic. The authors paid special attention to the entire molecular profile and emphasize that the comprehensive NGS analysis can reveal co-existing molecular events, which may have an important influence on the prognosis and response to the treatment. The molecular events present up-front can act as intristic resistance and those appearing under the treatment constitute the acquired resistance. Better understanding the intratumoral cellular heterogeneity may lead to identify the subgroups of genomic defined NSCLC that differs in their molecular profile. Therefore, it seems that it is no longer sufficient at settle the diagnosis of e.g., ALK-rearranged NSCLC, because new data informing of tumor heterogeneity can further enrich the molecular profile and have predictive value. Moreover, they can be basis for modifying the current treatment regimens to e.g., combine two different TKIs. Starting with presenting heterogeneity on the phenotype level they authors are further discussing heterogeneity in three genomic defined subtypes of NSCLC: KRAS- and EGFR-mutated and ALK-rearranged. The article contributes to the current knowledge about the importance of heterogeneity not only in preclinical setting, but the data of intratumoral heterogeneity are now entering in clinics and may guide the therapeutic decision. Please find attached the detailed review with questions/comments and minor errors.
Line 37 – There are some special metastatic sites, where the genetic differences are especially expressed, e.g., brain metastases differ from the matching primary tumor (e.g., Nicos M et al. Genomic Landscapes of DNA Copy Number Alterations in Primary Lung Cancers and Matched Brain Metastases P2.01-66, WCLC 2019, and Li L-L. et al. Mutational Heterogeneity between Primary Pulmonary Cancer Lesions and Matched Brain Metastases. WCLC 2019, P1.01-28)
Line 48 - 10-12 months.
However, for many tumors, e.g., ALK-positive NSCLC it is not true, where progression can appear after a couple of years (data from ALEX-study). It depends also on several factors: e.g., the treatment line, longer PFS are observed in treatment naïve patients than heavy pretreated. Moreover, generation of TKI used, where longer PFS are observed with higher generations of EGFR-TKI, ALK-TKI, etc.
Line 73: it is missing: squamous carcinomas between adenosquamous carcinomas, and sarcomatoid.
Line 81: Please specify: mutations and amplifications in EGFR, FGFR, PI3KCA, mutation in PTEN, Rb1, CDKN2A and DDR2 and amplification in MET are reported ( Soldera, S.V., Leighl, N.B., 2017. Update on the Treatment of Metastatic Squamous Non-Small Cell Lung Cancer in New Era of Personalized Medicine. Frontiers in Oncology 7. doi:10.3389/fonc.2017.00050)
Line 104- figure 1 gives a good illustration of the heterogeneity.
Line 129 – JAK-STAT pathway and rephrase the sentence as unclear: the immune negatively has negatively affected (what?)
Line 153 – It will be relevant to mention at least Sotorasib (AMG510). There are different grip points for targeting KRAS depending on the entire genomic profile (Salgia, R., Pharaon, R., Mambetsariev, I., Nam, A., Sattler, M., 2021. The improbable targeted therapy: KRAS as an emerging target in non-small cell lung cancer (NSCLC). Cell Reports Medicine 2, 100186. doi:10.1016/j.xcrm.2020.100186)
Line 159 – in this unit you may first underline that the EGFR heterogeneity can be considered in terms of intristic and acquired resistance and subsequently discuss both. It will make a better structured insight. (e.g., Santoni-Rugiu E et al. Intrinsic resistance to EGFR-Tyrosine Kinase Inhibitors in EGFR-Mutant Non-Small Cell Lung Cancer: Differences and Similarities with Acquired Resistance. Cancers, 2019 11, 923. doi: 10.3390/cancers11070923)
Line 195/Table 2 – please divide EGFR dominant and non-dominant by a horizontal line for make it clearer.
Line 226 – please rephrase the sentence since it is unclear for readers: Preclinical findings suggest the down regulation of ALK, STAT3, ERK and AKT phosphorylation 227 following EML4-ALK variant 1 inhibition.
The conclusion is not clear which variants are more sensitive to ALK-TKI. And elaborate a little more on this tread about why EML4-ALK variant 3 responds worse to treatment.
Line 261 - It is still unclear whether Brigatinib can overcome G1202R mutation, there are incoherent data: e.g., Sharma GG, Cortinovis D, Agustoni F, et al. A compound L1196M/G1202R ALK mutation in a patient with ALK-positive lung cancer with acquired resistance to Brigatinib also confers primary resistance to Lorlatinib. J Thorac Oncol.2019;14(11):e257–e259. doi:10.1016/j.jtho.2019.06.028 and Xiao, Z., Huang, X., Xie, B., Xie, W., Huang, M., Lin, L., 2020. Primary Resistance to Brigatinib in a Patient with Lung Adenocarcinoma Harboring ALK G1202R Mutation and LIPI-NTRK1 Rearrangement. OncoTargets and Therapy Volume 13, 4591–4595. doi:10.2147/ott.s249652.
Line 263/Table 3 - regarding ALK-non dominant mechanism: acquired BRAF mutation and phenotype change to EMT are missing. There are data illustrating the heterogeneity of acquired resistance under ALK-TKIs (e.g., Urbanska, EM. et al. Changing ALK-TKI-Resistance Mechanisms in Rebiopsies of ALK-Rearranged NSCLC: ALK- and BRAF-Mutations Followed by Epithelial-Mesenchymal Transition. International Journal of Molecular Sciences, 2020, 21, 2847. doi:10.3390/ijms21082847). You may consider rephrasing the subunit 6 and discuss the changing to different phenotypes as SCLC, squamous carcinoma and EMT.
Line 288 – develop the APOBEC acronym (The apolipoprotein B mRNA-editing enzyme, catalytic polypeptide-like)
Line 326 - on the condition however, that the patient belongs to the group of “shedders” and that there is an abundant amount of circulating DNA in blood.
Author Response
We thank the Reviewer for the positive feedback on our manuscript and revision suggestions. Please find our response below.
[1] Line 37 – There are some special metastatic sites, where the genetic differences are especially expressed, e.g., brain metastases differ from the matching primary tumor (e.g., Nicos M et al. Genomic Landscapes of DNA Copy Number Alterations in Primary Lung Cancers and Matched Brain Metastases P2.01-66, WCLC 2019, and Li L-L. et al. Mutational Heterogeneity between Primary Pulmonary Cancer Lesions and Matched Brain Metastases. WCLC 2019, P1.01-28)
- We thank the reviewer for the valuable suggestion. This information and references have now been added (line 40).
[2] Line 48 - 10-12 months.
However, for many tumors, e.g., ALK-positive NSCLC it is not true, where progression can appear after a couple of years (data from ALEX-study). It depends also on several factors: e.g., the treatment line, longer PFS are observed in treatment naïve patients than heavy pretreated. Moreover, generation of TKI used, where longer PFS are observed with higher generations of EGFR-TKI, ALK-TKI, etc.
- resolved
[3] Line 73: it is missing: squamous carcinomas between adenosquamous carcinomas, and sarcomatoid.
- resolved
[4] Line 81: Please specify: mutations and amplifications in EGFR, FGFR, PI3KCA, mutation in PTEN, Rb1, CDKN2A and DDR2 and amplification in MET are reported ( Soldera, S.V., Leighl, N.B., 2017. Update on the Treatment of Metastatic Squamous Non-Small Cell Lung Cancer in New Era of Personalized Medicine. Frontiers in Oncology 7. doi:10.3389/fonc.2017.00050)
- We thank the reviewer for the valuable suggestion. These information and references have now been added (line 85).
[5] Line 104- figure 1 gives a good illustration of the heterogeneity.
- figure 1 is already mentioned (line 97)
[6] Line 129 – JAK-STAT pathway and rephrase the sentence as unclear: the immune negatively has negatively affected (what?)
- We thank the reviewer for the suggestion. The sentence has now been rephrased (line 157)
[7] Line 153 – It will be relevant to mention at least Sotorasib (AMG510). There are different grip points for targeting KRAS depending on the entire genomic profile (Salgia, R., Pharaon, R., Mambetsariev, I., Nam, A., Sattler, M., 2021. The improbable targeted therapy: KRAS as an emerging target in non-small cell lung cancer (NSCLC). Cell Reports Medicine 2, 100186. doi:10.1016/j.xcrm.2020.100186)
- We thank the reviewer for the valuable suggestion. This information and references have now been added (line 182).
[8] Line 159 – in this unit you may first underline that the EGFR heterogeneity can be considered in terms of intristic and acquired resistance and subsequently discuss both. It will make a better structured insight. (e.g., Santoni-Rugiu E et al. Intrinsic resistance to EGFR-Tyrosine Kinase Inhibitors in EGFR-Mutant Non-Small Cell Lung Cancer: Differences and Similarities with Acquired Resistance. Cancers, 2019 11, 923. doi: 10.3390/cancers11070923)
- The suggested reference has been now mentioned (line 205)
[9] Line 195/Table 2 – please divide EGFR dominant and non-dominant by a horizontal line for make it clearer.
- resolved
[10] Line 226 – please rephrase the sentence since it is unclear for readers: Preclinical findings suggest the down regulation of ALK, STAT3, ERK and AKT phosphorylation 227 following EML4-ALK variant 1 inhibition.
The conclusion is not clear which variants are more sensitive to ALK-TKI. And elaborate a little more on this tread about why EML4-ALK variant 3 responds worse to treatment.
- We thank the reviewer for the suggestion. The sentence has now been rephrased (line 303).
[11] Line 261 - It is still unclear whether Brigatinib can overcome G1202R mutation, there are incoherent data: e.g., Sharma GG, Cortinovis D, Agustoni F, et al. A compound L1196M/G1202R ALK mutation in a patient with ALK-positive lung cancer with acquired resistance to Brigatinib also confers primary resistance to Lorlatinib. J Thorac Oncol.2019;14(11):e257–e259. doi:10.1016/j.jtho.2019.06.028 and Xiao, Z., Huang, X., Xie, B., Xie, W., Huang, M., Lin, L., 2020. Primary Resistance to Brigatinib in a Patient with Lung Adenocarcinoma Harboring ALK G1202R Mutation and LIPI-NTRK1 Rearrangement. OncoTargets and Therapy Volume 13, 4591–4595. doi:10.2147/ott.s249652.
- We agree with the reviewer’s comment and have added some more insights and the reference suggested (Line 346).
[12] Line 263/Table 3 - regarding ALK-non dominant mechanism: acquired BRAF mutation and phenotype change to EMT are missing. There are data illustrating the heterogeneity of acquired resistance under ALK-TKIs (e.g., Urbanska, EM. et al. Changing ALK-TKI-Resistance Mechanisms in Rebiopsies of ALK-Rearranged NSCLC: ALK- and BRAF-Mutations Followed by Epithelial-Mesenchymal Transition. International Journal of Molecular Sciences, 2020, 21, 2847. doi:10.3390/ijms21082847). You may consider rephrasing the subunit 6 and discuss the changing to different phenotypes as SCLC, squamous carcinoma and EMT.
- We thank the reviewer for the important reference suggestion, now discussed in the text (line 374)
[13] Line 288 – develop the APOBEC acronym (The apolipoprotein B mRNA-editing enzyme, catalytic polypeptide-like)
- resolved (line 384)
[14] Line 326 - on the condition, however, that the patient belongs to the group of “shedders” and that there is an abundant amount of circulating DNA in blood.
- The sentence has now been rephrased (line 438)